# Chicken Protein Hydrolysates Have Anti-Inflammatory Effects on High-Fat Diet Induced Obesity in Mice

**DOI:** 10.3390/medicines6010005

**Published:** 2018-12-28

**Authors:** Thomas A. Aloysius, Ana Karina Carvajal, Rasa Slizyte, Jon Skorve, Rolf K. Berge, Bodil Bjørndal

**Affiliations:** 1Department of Clinical Science, University of Bergen, N-5020 Bergen, Norway; thomas.aloysius@uib.no (T.A.A.); jon.skorve@uib.no (J.S.); rolf.berge@uib.no (R.K.B.); 2SINTEF Ocean, N-7465 Trondheim, Norway; ana.k.carvajal@sintef.no (A.K.C.); rasa.slizyte@sintef.no (R.S.); 3Department of Heart Disease, Haukeland University Hospital, N-5021 Bergen, Norway

**Keywords:** chicken protein hydrolysate, peptides, liver, obesity, mitochondrial fatty acid oxidation, plasma lipids, inflammation, cytokines

## Abstract

**Background**: Studies have shown that dietary source of protein and peptides can affect energy metabolism and influence obesity-associated diseases. This study aimed to investigate the impact of different chicken protein hydrolysates (CPHs) generated from chicken rest raw materials in a mouse obesity model. **Methods**: Male C57BL/6 mice were fed a high-fat, high-sucrose diet with casein or CPHs generated using Papain + Bromelain, Alcalase, Corolase PP, or Protamex for 12 weeks (*n* = 12). Body weight, feed intake, and intraperitoneal glucose tolerance was determined, and plasma and liver and adipose tissues were collected at sacrifice. **Results**: The average feed intake and body weight did not differ between the groups and white adipose tissue depots were unchanged, except for a reduction in the subcutaneous depot in mice fed the Protamex CPH diet. Moreover, the CPH diets did not prevent increased fasting glucose and insulin levels. Interestingly, the hepatic mitochondrial fatty acid β-oxidation was increased in mice fed Alcalase and Corolase PP CPHs. All CPH diets reduced plasma interleukine (IL)-1β, interferon-γ, tumor necrosis factor α, and monocyte chemotactic protein 1 compared to control, indicating anti-inflammatory effects. In addition, Corolase PP and Protamex CPHs significantly reduced plasma levels of IL-1α, IL-2, IL-6, IL-10, and granulocyte macrophage colony-stimulating factor. **Conclusions**: CPH diets were not able to counteract obesity and glucose intolerance in a mouse obesity model, but strongly reduced inflammatory parameters associated with obesity. Alcalase and Corolase PP CPHs also stimulated mitochondrial fatty acid β-oxidation. The possibility that hydrolysates from chicken rest raw materials could alleviate obesity-associated metabolic disease should be investigated further.

## 1. Introduction

Obesity and obesity-associated diseases have dramatically increased worldwide and pose a great challenge to human health [1]. The metabolic syndrome, consisting of abdominal obesity, elevated plasma triacylglycerol (TAG) and low high-density lipoprotein (HDL)-cholesterol levels, systemic inflammation, and glucose intolerance/insulin resistance, is highly linked to risk of cardiovascular diseases [2]. Moreover, diabetes mellitus type II [1] and non-alcoholic fatty liver disease [2] are among the metabolic diseases associated with obesity. Inflammation of metabolic tissues, particularly adipose tissue, is either triggered or exacerbated by nutritional overload and results in a low-grade chronic inflammatory state that will enhance the development of metabolic disturbances [3]. The adipose tissue is not only an organ for storage of fat, it is also metabolically active, and secretes a variety of hormones regulating energy metabolism [4,5]. During development of obesity, macrophages infiltrate the adipose tissue where they play a major role in cytokine production and activation of inflammatory pathways [6]. Cytokines produced by proinflammatory M1 macrophages are believed to contribute to the development of insulin resistance [7]. The cytokine tumor necrosis factor alpha (TNF-α) in adipocytes is positively correlated with the degree of obesity and hyperinsulinemia [8]. TNF-α has been linked to obesity-related insulin resistance, as it inhibits intracellular signaling from the insulin receptor and is considered a major factor in the development of metabolic diseases [9,10].

Mitochondrial function is highly linked to metabolic diseases, since increased oxidative stress due to nutritional overload will lead to mitochondrial dysfunction, further increasing the cardiovascular risk associated with obesity [11]. It has been postulated that dietary components such as omega-3 fatty acids can target mitochondrial function [8]. Thus, there is a focus on identifying nutritional approaches to prevent obesity and metabolic disease [12].

The activity of bioactive peptides has been a focus of intense research in the last years. Small peptides in the range of 2–12 amino acids are believed to cross the intestinal wall undigested and enter the circulation, where they can have a range of activities based on their amino acid composition and structure [13,14]. Natural peptides isolated from a number of food proteins, including plants and marine organisms, have been shown to have antioxidant effects in in vitro tests [15,16,17], and anti-hypertensive effects in mice through inhibition of the angiotensin-converting-enzyme (ACE) [18]. Protein hydrolysates from both animal- and plant-based sources can have specific effects based on their composition, as they will consist of a mixture of different, potentially bioactive, peptides. A number of studies have shown that egg- and soy-derived hydrolysates have anti-diabetic and anti-obesity properties in rodents, as recently reviewed by de Campos Zani et el. [19]. Our previous study in mice demonstrated that salmon protein hydrolysates generated by different enzymatic process conditions had distinctive effects on energy metabolism [20]. While one of the hydrolysates had no beneficial metabolic effect, two of the hydrolysates demonstrated anti-obesogenic and plasma TAG-lowering effects linked to reduced lipogenesis [20]. One of these salmon protein hydrolysates was also shown to have anti-inflammatory effects, as it lowered plasma cytokine levels and aortic plaque levels in an ApoE^-/-^ atherosclerosis mouse model [21].

Although the recent research has mainly focused on soy, milk, and marine protein hydrolysates, bioactive peptide mixtures could be generated from a wide range of protein sources. Chicken fileting gives rise to a large amount of rest raw material with high protein quality. In this study, we used a mouse model of diet-induced obesity to compare chicken protein hydrolysates (CPHs) produced using different commercially available proteases and their mixtures.

## 2. Materials and Methods

### 2.1. Animals and Diets

The animal study was conducted in accordance with the Guidelines for the Care and Use of Experimental Animals. The protocol was approved by the Norwegian Food Safety Authority (Protocol No. 6526). Date of approval: 9 May 2014. Male C57BL/6JBomTac mice, 8–10 weeks old, were purchased from Taconic (Ry, Denmark). Upon arrival, the mice were randomly placed in open cages, four in each cage, and they were allowed to acclimatize to their surroundings for one week before start of the experiment. The mice were kept in a 12 h light/dark cycle at a constant temperature (22 ± 2 °C) and a relative humidity of 55% (±5%). During the acclimatization period, the mice had unrestricted access to chow and tap water. The cages were block-randomized to five different diets, 12 mice per diet, equally distributed between four days of feeding start/sacrifice: Group 1 (control) was fed a high-fat diet consisting of 59 E% fat from lard, 19 E% protein from casein, and 22 E% from carbohydrates (Table 1). All ingredients were from Dyet Inc. (Bethlehem, PA, USA), except tert-butyl-hydroquinone, which was purchased from Sigma Aldrich (St. Louis, MO, USA). Groups 2–5 were fed high-fat diets with 9.5 E% casein and 9.5 E% CPHs generated with the enzymes Papain + Bromelain, Alcalase, Corolase PP, or Protamex, respectively. The hydrolysis was carried out in a 4-L closed glass vessel placed in a water bath (50 ± 2 °C). A marine impeller (approx. 50–60 rpm) was used in order to ensure homogeneity of the mixture during the entire hydrolysis. Rest raw materials (RRMs) from mechanical deboning of chicken meat (Nortura AS Hærland, Eidsberg, Norway) was used for the hydrolysis experiments. Four hydrolysis experiments were carried out on freshly minced chicken RRMs by adding commercially available enzymes or their mixtures. The experiment was performed by mixing minced chicken raw material with preheated water (50 °C) at the ratio 1:1. The enzymatic hydrolysis was started when the temperature of the mixture was 50 °C by adding either 0.1 % (by wet weight of raw material) of Alcalase^®^, Protamex^®^, Corolase^®^PP, or 0.05% + 0.05% of a mixture of Papain and Bromelain, respectively (1:1 *w*/*w*). The hydrolysis proceeded for 60 min Hydrolyses was terminated by enzyme inactivation by microwave heating for 5 min at a temperature higher than 90 °C. The mixtures were centrifuged and chicken protein hydrolysate (CPH, water-soluble compounds) was freeze-dried and used for feeding tests.

Mice were fed ad libitum for 12 weeks. At the end of the study, mice were anaesthetized by inhalation of 2–5% isoflurane (Schering-Plough, Kent, UK). The abdomen was opened in the midline and EDTA-blood was collected by cardiac puncture of the right ventricle and immediately chilled on ice. Blood samples were centrifuged and plasma was stored at −80 °C prior to analysis. Liver weight was determined, and one liver lobe was instantly flash frozen using a clamp immersed in liquid nitrogen and stored at −80 °C for further analysis. The right side subcutaneous-, gonadal-, and kidney- (perirenal and retroperitoneal) white adipose tissue depots (WAT) and the mesenteric WAT were dissected and weighed according to Johnson and Hirsch [22] (with commentary in: http://bnorc.org/cores/protocols/dissectionx/).

### 2.2. Intraperitoneal Glucose Tolerance Test and Insulin Analysis

At baseline and week 11, six mice per group were fasted for four hours, and maximum 200 µL EDTA-blood was drawn from the saphenous vein of the hind leg, centrifuged for 15 min at 3000× *g*, and plasma was stored at −80 °C prior to analysis. The fasting blood glucose level was measured using FreeStyle Lite (Abbot Diabetes Care Inc., Alameda, CA, USA). Mice were then injected with an intraperitoneal dose of 20% glucose in phosphate buffered saline corresponding to 2 g glucose/kg body weight. Blood glucose levels were measured at 15, 30, 60, and 120 min after the intraperitoneal injection. Insulin was measured in plasma using the rat/mouse insulin ELISA kit from Merck (Kenilworth, NJ, USA).

### 2.3. Plasma Lipids

Plasma lipids were measured enzymatically on a Hitachi 917 system (Roche Diagnostics GmbH, Mannheim, Germany) using the triacylglycerol (GPO-PAP) and cholesterol kit (CHOD-PAP) from Roche Diagnostics, and the phospholipids FS kit and NEFA FS kit from DiaSys (Diagnostic Systems GmbH, Holzheim, Germany), as previously described [23].

### 2.4. Liver Enzyme Activities

One gram of fresh liver sample was chilled on ice and homogenized in 4 mL ice-cold sucrose medium (0.25 M sucrose, 10 mM HEPES, and 1 mM Na_4_EDTA, adjusted to a pH of 7.4 with KOH/HCl). The homogenates were centrifuged at 1030 RCF for 10 min at 4 °C and the post-nuclear fraction was removed and used for further analysis [24]. Palmitoyl-CoA oxidation was measured immediately in the post-nuclear fraction from fresh liver as acid-soluble products, as previously described [25], with some modifications [26]. Acyl-CoA oxidase (ACOX1; EC: 1.3.3.6) was measured in frozen post-nuclear fraction as previously described [27].

### 2.5. Hepatic Gene Expression

Total cellular RNA was purified from 20 mg snap-frozen liver samples, and 1000 ng RNA was reverse transcribed using High Capacity cDNA Reverse Transcription Kits (Applied Biosystems, Foster City, CA, USA). Real-time PCR was performed with Sarstedt 384-well Multiply-PCR plates (Sarstedt Inc., Newton, NC, USA) on the following genes, using probes and primers from Applied Biosystems: Carnitine palmitoyl transferase 1a and 2 (*Cpt1a,* Mm01231183_m1, and *Cpt2*, Mm00487205_m1), hydroxyacyl-CoA dehydrogenase trifunctional multienzyme complex subunit beta (*Hadhb*, Mm00523880_g1), and 2,4-dienoyl-CoA reductase 1 (*Decr1*, Mm00470689_m1). Each gene was run with a standard curve using cDNA from QPCR mouse reference RNA (Agilent Technologies, Santa Clara, CA, USA). Results normalized to Eukaryotic 18S ribosomal RNA (*18S*, Kit-FAM-TAMRA (Reference RT-CKFT-18s) from Eurogentec, Liege, Belgium) are presented.

### 2.6. Plasma Inflammatory Markers

Plasma concentrations of interleukin (IL)-1α, IL-1β, IL-2, IL-6, IL-10, IL-17, granulocyte colony-stimulating factor (G-CFS), granulocyte macrophage colony-stimulating factor (GM-CFS), interferon gamma (IFN-γ), monocyte chemotactic protein 1 (MCP-1/CCL2), chemokine (C-C motif) ligand 5 (RANTES/CCL5), and TNF-α were determined using a custom-made multiplex MILLIPLEX MAP kit (Millipore Corp., St. Charles, IL, USA). The antibody-conjugated beads were allowed to react with the sample and a secondary antibody in a 96-well plate to form a capture sandwich immunoassay. Finally, the assay solution was read by the Bio-Plex array reader (Bio-Rad, Hercules, CA, USA) and determined with the Bio-Plex Manager Software 4.1 (Bio-Rad).

### 2.7. Statistical Analysis

Data sets were analyzed using Prism Software 6.0 (Graph-Pad Software, San Diego, CA, USA). The results are shown as means of 12 animals per group with their standard deviations, unless otherwise stated. One-way ANOVA with Fisher’s least significant difference (LSD) post hoc test was used to evaluate statistical differences between control and treatment groups. Two-way ANOVA with Sidak’s multiple comparisons test was used to evaluate statistical differences within each group at different time points. In this study, *p*-values < 0.05 were considered statistically significant.

## 3. Results

### 3.1. Weight Gain in C57BL/6 Mice after 12 Weeks on an Obesogenic Diet

Mice were fed high-fat/high-sucrose diets with either casein as the only protein source or diets with a combination of 9.5 E% from casein and 9.5 E% from CPHs generated using four different enzymes (Papain + Bromelain, Alcalase, Corolase PP, or Protamex). Body weight measured at different time points during the study did not reveal differences between the feeding groups (Figure 1A), and the total weight gain was similar in all groups (Figure 1B). The feed intake was measured during week 1, week 6, and week 11 of the study. The average feed intake per mouse did not differ between groups (data not shown), despite a higher feed intake during the first week of the study in some CPH groups (Figure 1C).

Weights of four different white adipose tissue (WAT) depots were determined at sacrifice. The subcutaneous WAT depot on the right side of the body from fore- to hindlimb, the right side epididymal/gonadal WAT depots, the mesenteric WAT, and the right-side WAT depot surrounding the kidney. The WAT depots of mice fed CPH diets were not different in weight compared to casein-fed mice, except a small but significant reduction in the subcutaneous WAT depot of mice fed the Papain + Bromelain- and Protamex CPH diet (Figure 2A–D). This led to a significantly higher ratio between visceral and subcutaneous WAT depots in the Protamex CPH-fed mice (Figure 2E).

### 3.2. Intraperitoneal Glucose Tolerance Test

To determine if the obesogenic diet gave obesity-associated metabolic disturbances, an intraperitoneal glucose tolerance test (IP-GTT) was performed at baseline and towards the end of the study. At baseline, mice had a normal fasting glucose level, but after 11 weeks of high-fat/high-sucrose diets, fasting glucose levels increased by 15–30% in all groups (Figure 3A). Fasting insulin levels were also substantially increased by the obesogenic diet, resulting in between 4- to 10-fold higher values at 11 weeks compared to baseline (Figure 3B). This led to a strong reduction in the glucose/insulin ratio at 11 weeks compared to baseline (Figure 3C). In addition to higher fasting glucose levels, the increase in glucose after a 2 g/kg body weight glucose injection, and the time required for glucose-levels to return to fasting levels, was highly increased after 11 weeks of diet compared to baseline (Figure 4A,B). The delta area under the curve (AUC) increased in all groups from baseline to 11 weeks, with no significant difference between diets (Figure 4C). Thus, obesity-associated disturbances in glucose metabolism were observed, but the CPH diets did not influence these parameters compared to the casein diet.

### 3.3. Hepatic Fatty Acid Oxidation and Plasma Lipid Levels

In order to determine if the CPH diets could influence hepatic fatty acid oxidation, in vitro β-oxidation of palmitoyl-CoA was measured in fresh hepatic liver homogenates. Both the Alcalase CPH and the Corolase PP CPH significantly increased β-oxidation compared to casein (by 32% and 35%, respectively; Figure 5A). The observed increase in β-oxidation seemed to be due to mitochondrial and not peroxisomal fatty acid degradation, since the activity of ACOX1, the rate-limiting enzyme in peroxisomal fatty acid oxidation, was unaltered (Figure 5B). Genes involved in mitochondrial import of fatty acids (*Cpt1a* and *Cpt2*), and subsequent β-oxidation (*Hadhb* and *Decr1*) were, however, not influenced by the CPH diets (Figure 5C).

The increase in hepatic mitochondrial β-oxidation compared to control in mice fed some of the CPH diets did not result in reduced plasma lipid levels, as plasma TAG was significantly increased in Papain + Bromelain and Alcalase CPH diets, while total cholesterol, phospholipid, and non-esterified fatty acids (NEFA) levels were unchanged (Figure 6A–D). Hepatic TAG, cholesterol, and phospholipids were not affected by the CPH diets (Figure 6E–G), but NEFA was significantly reduced by Papain + Bromelain, Alcalase, and Corolase PP CPH diets (Figure 6H).

### 3.4. The Effect of Chicken Protein Hydrolysate Diets on Plasma Cytokine Levels

Inflammation is an integral component of obesity-associated diseases, and we wished to investigate if CHP diets could influence plasma pro-inflammatory cytokine levels. Interestingly, all CPHs showed a clear anti-inflammatory potential in that they significantly reduced plasma IL-1β, IFN-γ, TNF-α, and MCP-1 compared to the casein control diet (Figure 7). In addition, the Papain + Bromelain CPH reduced G-CFS; Alcalase CPH reduced IL-2, IL-6, and RANTES; Corolase PP reduced IL-1α, IL-2, IL-6, and GM-CSF; while the Protamex CPH reduced IL-1α, IL-2, IL-6, IL-17, and GM-CSF. Surprisingly, the anti-inflammatory cytokine IL-10 was reduced by Papain-Bromelain CPH, Corolase PP CPH, and Protamex CPH diets. Altogether, CPHs generated using Corolase PP and Protamaex seemed to have the strongest cytokine-lowering potential.

## 4. Discussion

In this obesity model in male C57BL/6 mice, diets with different chicken protein hydrolysates constituting 50% of the protein source were able to significantly reduce plasma cytokine levels compared to a casein diet. Two of the hydrolysates, generated using Alcalase and Corolase PP, also stimulated hepatic mitochondrial fatty acid oxidation, but none of the CPH diets had an effect on weight gain.

The high-fat diet used to induce obesity in this study was based on standard commercial diets using 19 E% from protein, 59 E% from fat, and 22 E% from carbohydrates, with a high proportion of disaccharides. C57BL/6 mice have been established as a good model of diet-induced obesity, as these mice are lean and healthy on a low-fat diet, but develop insulin resistance, hyperglycemia, and obesity on a high-fat diet [28,29]. A minimum of 12 weeks of feeding is required to develop obesity in rodent models. Although we did not have a low-fat control group in our experiment, the body weight and fat-depot size of the mice after 12 weeks were indicative of obesity [28]. Mice fed the Protamex CPH diet had a significantly lower level of subcutaneous adipose tissue than casein-fed mice, but the ratio between visceral and subcutaneous adipose tissue was not improved. As the visceral adipose has high hormonal activity compared to subcutaneous adipose tissue, and has been directly linked to pathological conditions including insulin resistance [30], a beneficial health effect of subcutaneous adipose tissue reduction is not clear.

We demonstrated a dramatic increase in fasting glucose and insulin levels after 11 weeks of high-fat feeding (Figure 3), and a reduced blood glucose clearance after an intraperitoneal injection of glucose (Figure 4). The ratio between fasting glucose and insulin is a good measure of insulin resistance, and in humans a value below 4–5 is considered abnormal [27]. Thus, the mice developed obesity and the expected symptoms of insulin resistance, and adding chicken hydrolysates to the diet did not prevent weight gain or elevated plasma glucose and insulin levels. Marine hydrolysates have been shown to reduce weight gain in feeding studies using 35–45 E% from fat, including Wistar rats [31] and C57BL/6 mice [20,32] in short-term studies, and LDLR-/-/ApoB100/100 mice [33] in a long-term study. Few studies have documented the prevention of obesity in long-term high-fat feeding studies in mice, with some exceptions; for instance, oral supplementation of hydrolysate from velvet antler had potent anti-obesity effects [34]. It is possible that very potent peptides are necessary to prevent diet-induced obesity in ≥60 E% fat-diets and that effect on weight gain by CPHs could have been seen with a high-fat diet with lower fat levels. Similarly, few studies have shown effects of hydrolysates on glucose metabolism in ≥60 E% fat diet-induced obesity models in rodents. However, a number of studies in genetic animal models of obesity show that hydrolysates can potentially affect glucose tolerance. For example, Dong et al. demonstrated that blood glucose levels and the serum insulin sensitivity index was reduced after 10 weeks treatment of db/db mice with *Apostichopus japonicus* hydrolysates [35], while whey protein hydrolysate was shown to improve glucose clearance after intraperitoneal glucose injection in ob/ob mice and also reduced fasting insulin levels without affecting fasting glucose levels [36].

Increased inflammation, believed to particularly originate from visceral adipose tissue, is a key component in obesity-associated metabolic disturbances [5]. Inflammation and macrophage infiltration appear to be more distinct in the visceral adipose tissue than in the subcutaneous adipose tissue [37,38]. Similar to hydrolysates from casein [39], all the CPHs tested had potent cytokine-lowering effects in plasma compared to the casein control group after 12 weeks of feeding. Some variation in the anti-inflammatory potential of the differently generated CPHs was observed, as the hydrolysates produced by using the enzymes Corolase PP and Protamex lowered more cytokines than the hydrolysates produced by using Papain + Bromelain or Alcalase. A reduction in the known anti-inflammatory cytokine IL-10 was associated with the CPH-induced lowering of plasma cytokines and chemokines. This is similar to previous studies on marine hydrolysates, where a salmon protein hydrolysate lowered atherosclerotic plaque formation in ApoE-/- mice and reduced a number of cytokines and chemokines, including IL-10 [40]. Thus, protein hydrolysates may potentially inhibit the release of cytokines and chemokines, including anti-inflammatory cytokines, and this should be investigated further.

As well-functioning mitochondria are important to prevent obesity-associated metabolic disturbances, we screened the mice livers for β-oxidation activity after 12 weeks of high-fat feeding with or without CPH in the diet. Interestingly, we found that both the Alcalase-generated CPH and the Corolase PP-generated CPH significantly increased in vitro palmitoyl-CoA degradation in fresh liver homogenates with intact mitochondria, but did not increase peroxisomal ACOX1 activity. This shows that mitochondrial function could be influenced by the CPHs, although hepatic genes involved in mitochondrial β-oxidation, *Cpt1a*, *Cpt2*, *Hadhb*, and *Decr1*, were not affected. As these genes are peroxisome proliferator-activated receptor alpha (PPARα) response genes, the increase in β-oxidation observed in Alcalase and Corolase PP CPH-fed mice did not seem to be through PPARα activation mechanisms. Altogether, our results indicate a moderate effect of Alcalase and Corolase PP CPHs on lipid catabolism in this obesogenic mouse model.

Lowering of plasma TAG has previously been observed with salmon protein hydrolysates in mice fed a moderate high-fat diet, and this reduction was linked to reduced expression of genes involved in lipogenesis, with minor effects on mitochondrial β-oxidation [20]. Here, the increase in β-oxidation activity was not linked to a reduction of plasma or hepatic TAG levels. In fact, plasma TAG levels were significantly increased in the Papain + Bromelain CPH and the Alcalase CPH group compared to the control group. Interestingly, we previously observed that one of three salmon hydrolysates generated through different enzymatic procedures increased plasma TAG levels [20], independent of hepatic β-oxidation activity or feed intake. Feed intake was only measured at three time points during the current study (for one week in the beginning, middle, and end of the study), and it is difficult to conclude if the high TAG level was caused by differences in feed intake. Significant differences in feed intake were observed during the first week of feeding, indicating a difference in palatability. Also, mice were not fasted before they were killed, and this may have affected the TAG levels. Previous studies have shown that activation of mitochondrial biogenesis and enhanced mitochondrial fatty acid oxidation can prevent diet-induced obesity [41], and thus a small increase in activity could still be considered beneficial. Interestingly, the hepatic level of NEFA was reduced in the Papain + Bromelain, Alcalase, and Corolase PP CPH groups, which could indicate a reduction in potentially toxic lipid intermediates. Importantly, the anti-inflammatory effects of CPHs could be important for maintaining efficient mitochondrial function, as diet-induced inflammation is a contributor to mitochondrial abnormalities [42]. The lack of a low fat-fed control group is a limitation of the study, and although the cytokine-lowering potential of CPHs was evident, we cannot conclude whether plasma inflammation was returned to normal levels.

## 5. Conclusions

Altogether, our findings show that CPH diets were not able to counteract obesity and glucose intolerance in a mouse obesity model, but strongly reduced inflammatory parameters associated with obesity. In addition, Alcalase and Corolase PP CPHs stimulated mitochondrial β-oxidation. This indicates that hydrolysates from chicken rest raw materials are able to stimulate mitochondrial function and reduce obesity-associated inflammation. Their ability to alleviate obesity-associated metabolic disease should be further investigated.

## Figures and Tables

**Figure 1 medicines-06-00005-f001:**
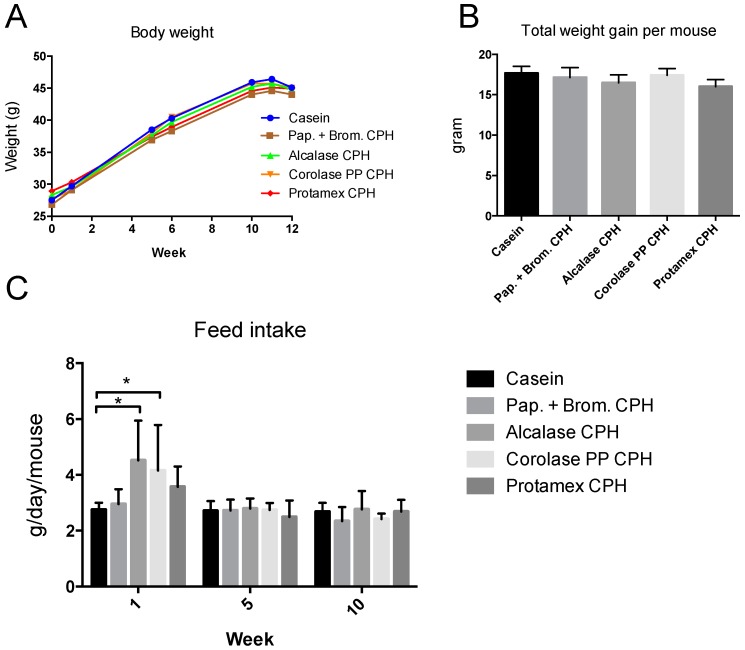
Body weight and feed intake in C57BL/6 mice fed 19 E% casein- or 9.5 E% casein and 9.5 E% CPH-high fat/high sucrose diets for 12 weeks. (**A**) Body weight per mouse measured before and after week 1, 6, and 11, and at day of sacrifice. Mean values are shown (*n* = 12). (**B**) Total weight gain per mouse during the study. Mean values with standard deviations are shown (*n* = 12). (**C**) Feed intake per week per mouse measured during week 1, 6, and 11 of the study. Mean values were based on average feed intake per mouse per cage (*n* = 3). Significant difference between values were determined by one-way ANOVA with Fisher’s LSD test (* *p* < 0.05).

**Figure 2 medicines-06-00005-f002:**
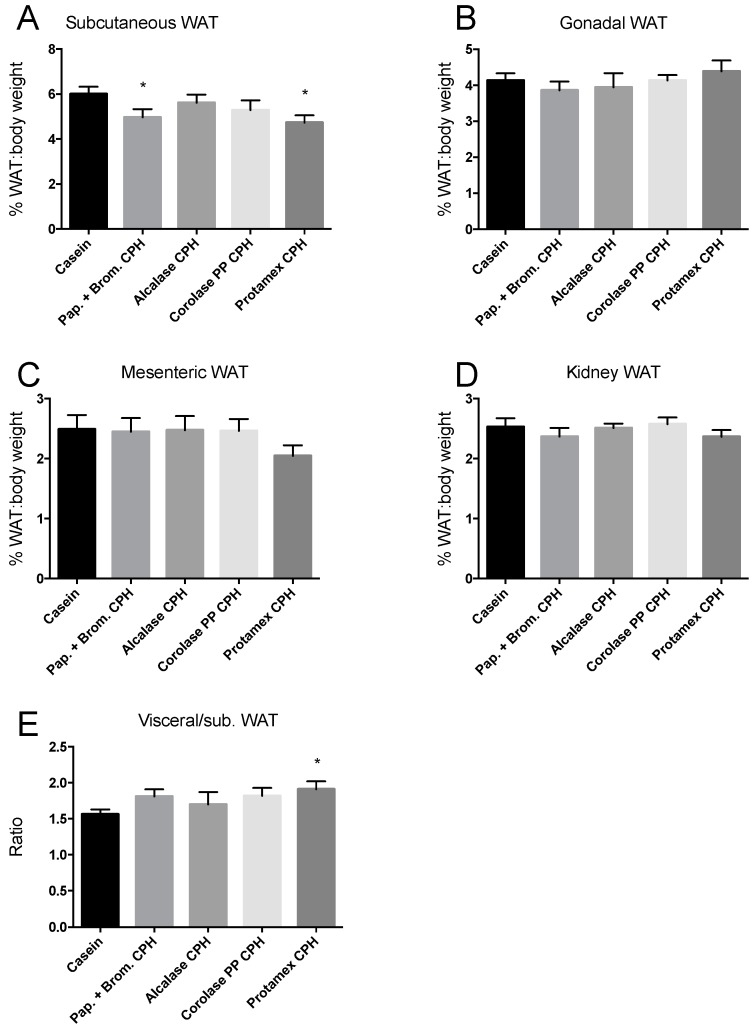
Weight percentage of body weight of white adipose tissue (WAT) depots in C57BL/6 mice fed 19 E% casein- or 9.5 E% casein and 9.5 E% CPH-high fat/high sucrose diets for 12 weeks. (**A**) The subcutaneous WAT depot on the right side of the body from fore- to hindlimb × 2. (**B**) The right side epididymal/gonadal WAT depot × 2. (**C**) The mesenteric WAT surrounding the intestines. (**D**) The right-side WAT depot surrounding and behind the kidney × 2. (**E**) The ratio between the weight percentage of body weight of visceral and subcutaneous WAT depots. Mean values with standard deviation are shown (*n* = 11–12). Significant difference between values were determined by one-way ANOVA with Fisher’s LSD test (* *p* < 0.05).

**Figure 3 medicines-06-00005-f003:**
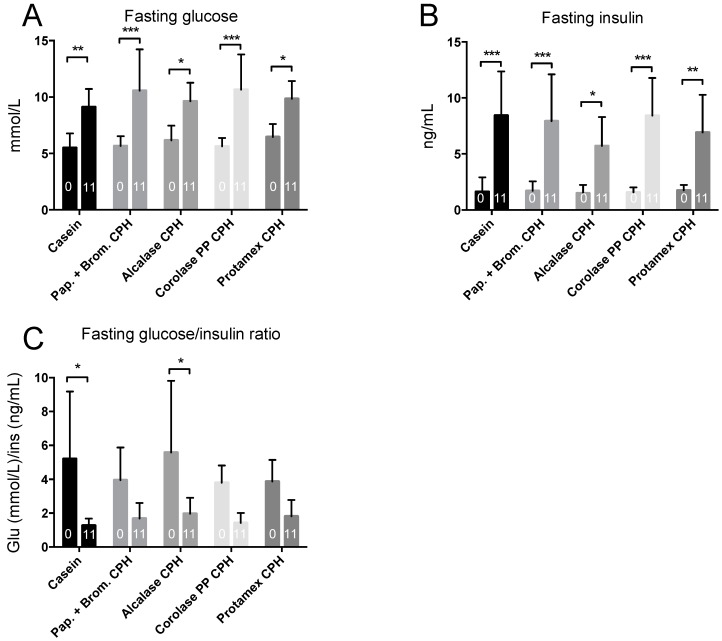
Fasting glucose and insulin levels in C57BL/6 mice at baseline (0 weeks) and after 11 weeks of feeding with 19 E% casein- or 9.5 E% casein and 9.5 E% CPH-high fat/high-sucrose diets. (**A**) The fasting blood glucose level. (**B**) The fasting plasma insulin level. (**C**) The fasting glucose and insulin ratio. Mean values with standard deviation are shown (*n* = 5–6). Significant difference between group mean values at 0 and 11 weeks was determined by one-way ANOVA with Fisher’s LSD test (all *p* > 0.05). Significant difference between mean values at 0 and 11 weeks within each group was determined by two-way ANOVA and Sidak’s multiple comparisons test (* *p* < 0.05, ** *p* < 0.01, *** *p* < 0.001).

**Figure 4 medicines-06-00005-f004:**
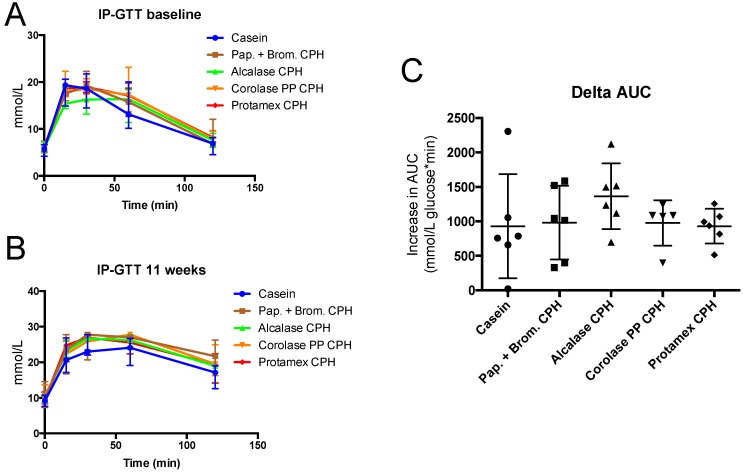
Intraperitoneal glucose tolerance (IP-GTT) in C57BL/6 mice at baseline (**A**) and after 11 weeks of feeding with 19 E% casein- or 9.5 E% casein and 9.5 E% CPH-high fat/high-sucrose diets (**B**). The increase in the area under the curve (AUC) from baseline to 11 weeks of feeding (**C**). Values are means with standard deviation (*n* = 5–6). Significant difference between values were determined by one-way ANOVA with Fisher’s LSD test (*p* < 0.05).

**Figure 5 medicines-06-00005-f005:**
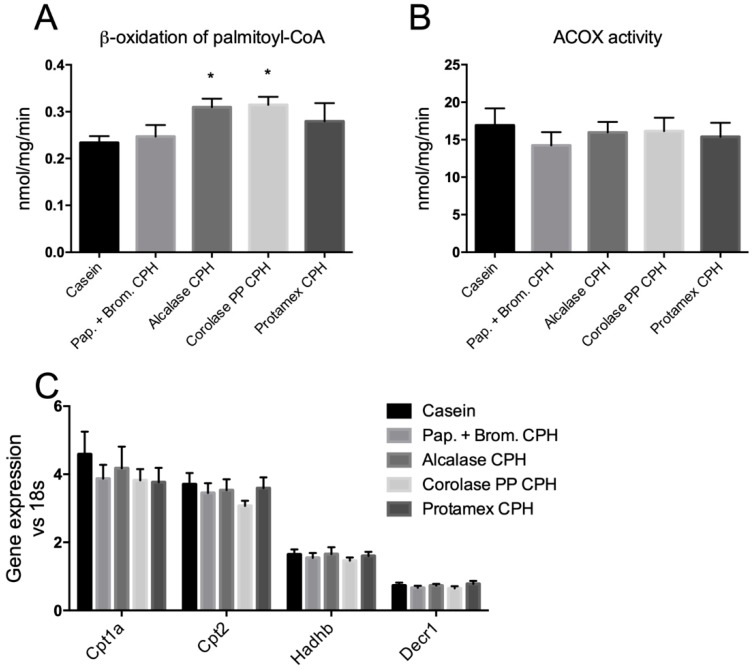
Hepatic β-oxidation in C57BL/6 mice fed high fat/high sucrose diets with casein (19 E%) or casein (9.5 E%) combined with different CPHs (9.5 E%) for 12 weeks. (**A**) In vitro β-oxidation of palmitoyl-CoA in fresh liver homogenates. (**B**) In vitro activity of acyl-CoA oxidase 1 (ACOX1) in frozen liver homogenates. Mean values with standard deviation are shown (*n* = 5–6). (**C**) Hepatic gene expression of carnitine palmitoyl transferase 1a (*Cpt1a*), *Cpt2*, hydroxyacyl-CoA dehydrogenase trifunctional multienzyme complex subunit beta (*Hadhb*), and 2,4-dienoyl-CoA reductase 1 (*Decr1*). Mean values with standard deviation are shown (*n* = 7). Significant differences between values were determined by one-way ANOVA with Fisher’s LSD test (* *p* < 0.05).

**Figure 6 medicines-06-00005-f006:**
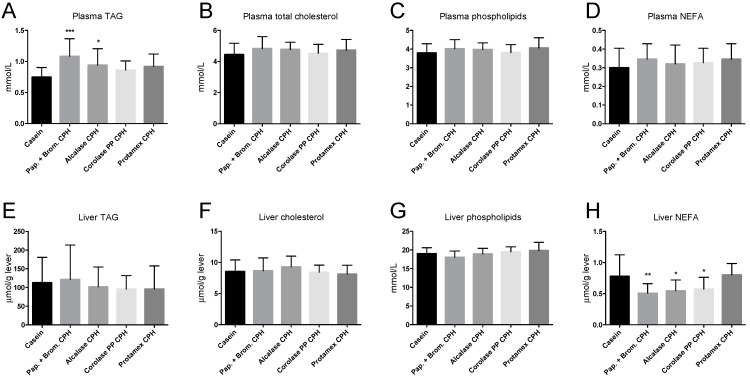
Plasma and liver lipid levels in C57BL/6 mice fed 19 E% casein- or 9.5 E% casein and 9.5 E% CPH-high fat/high sucrose diets for 12 weeks (**A**) Plasma triacylglycerol (TAG), (**B**), plasma total cholesterol, (**C**) plasma phospholipids, (**D**) plasma non-esterified fatty acids (NEFA), (**E**) liver TAG, (**F**) liver cholesterol, (**G**) liver phospholipids, (**H**) liver NEFA. Values are means with standard deviation (*n* = 11–12). Significant differences between values were determined by one-way ANOVA with Fisher’s LSD test (* *p* < 0.05, ** *p* < 0.01, *** *p* < 0.001).

**Figure 7 medicines-06-00005-f007:**
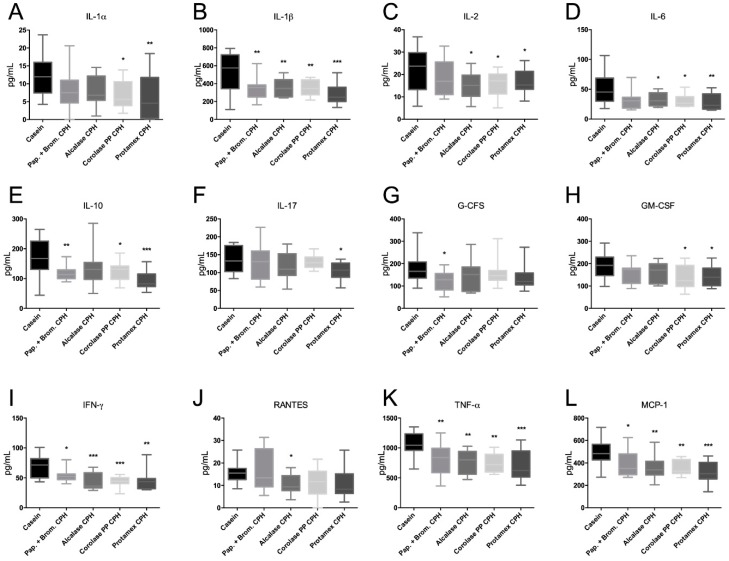
Plasma concentrations of (**A**) interleukin (IL)-1α, (**B**) IL-1β, (**C**) IL-2, (**D**) IL-6, (**E**) IL-10, (**F**) IL-17, (**G**) granulocyte colony-stimulating factor (G-CFS), (**H**) granulocyte macrophage colony-stimulating factor (GM-CFS), (**I**) interferon gamma (IFN-γ), **(J**) chemokine (C-C motif) ligand 5 (RANTES/CCL5), (**K**) tumor necrosis factor alpha (TNF-α), and (**L**) monocyte chemotactic protein 1 (MCP-1/CCL2),. Mean values with ranges are shown (*n* = 10–12). Significant differences between values were determined by one-way ANOVA with Fisher’s LSD test (* *p* < 0.05, ** *p* < 0.01, *** *p* < 0.001).

**Table 1 medicines-06-00005-t001:** Nutrient composition of the diets in energy % (E%) and weight (g).

Nutrients	Casein	Papain + Bromelain CPH	Alcalase CPH	Corolase PP CPH	Protamex CPH
Protein (E%)	19	19	19	19	19
Casein (g)	250	125	125	125	125
CPH (g)		125	125	125	125
Fat (E%)	59	59	59	59	59
Soy oil (g)	35	35	35	35	35
Lard (g)	315	315	315	315	315
Carbohydrate (E%)	22	22	22	22	22
Cornstarch (g)	100	100	100	100	100
Dyetrose (g)	100	100	100	100	100
Sucrose (g)	100	100	100	100	100
Fiber (g)	50	50	50	50	50
Micronutrients (E%)	0	0	0	0	0
AIN-93G-MX mineral mix (g)	34	34	34	34	34
AIN-93-VX vitamin mix (g)	10	10	10	10	10
L-Cysteine (g)	3	3	3	3	3
Choline bitartrate (g)	2.5	2.5	2.5	2.5	2.5
Tert-butyl-hydroquinone (g)	014	014	014	014	014

Abbreviation: CPH, chicken protein hydrolysate.

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
