# Peer review of "Chicken Protein Hydrolysates Have Anti-Inflammatory Effects on High-Fat Diet Induced Obesity in Mice"

_medicines, 2018, doi:10.3390/medicines6010005_

Round 1
Reviewer 1 Report
In this manuscript, the authors aimed to investigate the impact of different chicken protein hydrolysates (CPH) generated from chicken rest raw materials in a mouse obesity model. The summarizes part of the results are discussed in the paper as chicken protein hydrolysates (CPH) diets were not able to counteract obesity and glucose intolerance in a mouse obesity model, but potentially reduced inflammatory markers associated with obesity.
The manuscript is well written in terms of the English language. However, the manuscript has few typo errors and a number of formatting issue. The topic is of great interest for clinicians and Endocrinologist. The manuscript provides an ample introduction and sufficient background to understand the study. The results are clearly presented and conclusions supported by the study results.
Nonetheless, the manuscript required to improve by reviewing a few minor concerns.
(1) There are some issue of formatting in the paper as clearly seen from the PDF file of the paper. More gap are seen after full stop in many places.
(2) In the introduction part ‘Protein hydrolysates from both animal- and plant-based sources will consist of a mixture of 60 potentially bioactive peptides that can have specific effects based on their composition’ this line is grammatically wrong. Please update the line.
(3) Authors have to provide clearly about the protocol number or ethical approval number (IRB) in the manuscript.
(4) For the collection of white adipose tissue (WAT) depots in C57BL/6 mice, it is advised that the author should use picture to show the exact location of the collections of depots, I think pictures are better representation of this. If the authors have those picture kindly update your manuscript based on the picture observation.
(5) In the figure 5, the legend used near the photo was a typo error, please update it. In the same figure “c” the name of hepatic gene expression should be clearly understood.
(6) Please check the figure 7 also for IL-1α, interferon gamma and tumour necrosis factor alpha (Check the PDF version only).
(7) Please explain the mysterious finding of your study “the increase in beta oxidation activity was not linked to a reduction of plasma or hepatic TAG-levels. In fact, plasma TAG levels were significantly increased in the Papain + Bromelain CPH and the Alcalase CPH group compared to control”. Do you have any references for this, to support your study?
(8) Please update the full information of the cited papers in the references such as volume, issue number and page numbers.
- Anupama N.; Sindhu G.; Raghu K.G. Significance of mitochondria on cardiometabolic syndromes. Fundam Clin Pharmacol 2018.
-- Jensen I.J.; Maehre H.K. Preclinical and Clinical Studies on Antioxidative, Antihypertensive and Cardioprotective Effect of Marine Proteins and Peptides-A Review. Mar Drugs 2016, 14
-- Chai T.T.; Law Y.C.; Wong F.C.; Kim S.K. Enzyme-Assisted Discovery of Antioxidant Peptides from Edible Marine Invertebrates: A Review. Mar Drugs 2017, 15
de Campos Zani S.C.; Wu J.; Chan C.B. Egg and Soy-Derived Peptides and Hydrolysates: A Review of 400 Their Physiological Actions against Diabetes and Obesity. Nutrients 2018, 10
-- Vik R.; Parolina C.; Bjorndal B.; Busnelli A.; Holm S.; Brattelid T., et al. A salmon protein hydrolysate excerts lipid-independent anti-atherosclerotic activity in Apo E-deficient mice. Atherosclerosis 2014, 235, E188.
After amendment of the above comments in the manuscript, I would support in favor of the author for the publication of this paper.
Author Response
Reply to reviewers:
We greatly appreciate the valued input from the reviewer, and the opportunity to resubmit our manuscript"Chicken protein hydrolysates have anti-inflammatory effects on high-fat diet induced obesity in mice" (Man. no. Medicines-412488). We have attempted to address all the questions of the reviewer, and hope that this has resulted in an improved manuscript. All changes are marked in the revised manuscript.
Reviewer report and replies:
In this manuscript, the authorsaimed to investigate the impact of different chicken protein hydrolysates (CPH) generated from chicken rest raw materials in a mouse obesity model.The summarizes part of the results are discussed in the paper as chicken protein hydrolysates (CPH) diets were not able to counteract obesity and glucose intolerance in a mouse obesity model, but potentially reduced inflammatory makers associated with obesity.
The manuscript is well written in terms of the English language. However, the manuscript has few typo errors and a number of formatting issue. The topic is of great interest for clinicians and Endocrinologist.The manuscript provides an ample introduction and sufficient background to understand the study. The results are clearly presented and conclusions supported by the study results.
Nonetheless, the manuscript required to improve by reviewing a few minor concerns.
(1) There are some issue of formatting in the paper as clearly seen from the PDF file of the paper. More gap are seen after full stop in many places.
REPLY: We apologize for the formatting errors. All gaps have been removed, and the manuscript has been proof read to improve language and remove overlooked errors.
(2) In the introduction part ‘Protein hydrolysates from both animal- and plant-based sources will consist of a mixture of 60 potentially bioactive peptides that can have specific effects based on their composition’ this line is grammatically wrong. Please update the line.
REPLY: The sentence has been altered to make it grammatically correct: “Protein hydrolysates from both animal- and plant-based sources can have specific effects based on their composition, as they will consist of a mixture of different, potentially bioactive, peptides”
(3) Authors have to provide clearly about the protocol number or ethical approval number (IRB) in the manuscript.
REPLY: The animal experiment was approved by the Norwegian Food Safety Authority, and this has been corrected in the manuscript (Materials and Methods, Animals and diet, page 6): “The protocol was approved by the Norwegian Food Safety Authority (Protocol no. 6526).”
(4) For the collection of white adipose tissue (WAT) depots in C57BL/6 mice, it is advised that the author should use picture to show the exact location of the collections of depots, I think pictures are better representation of this. If the authors have those picture kindly update your manuscript based on the picture observation.
REPLY: We apologize for the lack of description of dissection of liver and adipose tissues. We unfortunately do not have pictures of the fat depots, but have included a description of the dissection procedure in Materials and Methods, page 8: “Liver weight was determined, and one liver lobe was instantly flash frozen using a clamp immersed in liquid nitrogen, and stored at -80°C for further analysis. The right side subcutaneous-, gonadal-, and kidney (perirenal and retroperitoneal) white adipose tissue depots (WAT) as well as the mesenteric WAT were dissected and weighed according to Johnson and Hirsch [22] with commentary in: http://bnorc.org/cores/protocols/dissectionx/.”
(5) In the figure 5, the legend used near the photo was a typo error, please update it. In the same figure “c” the name of hepatic gene expression should be clearly understood.
REPLY: The figure legend has been corrected, and the figure has been modified so that the gene names are hopefully more clearly readable.
(6) Please check the figure 7 also for IL-1α, interferon gamma and tumour necrosis factor alpha (Check the PDF version only).
REPLY: We have added a tiff-version of the figure which hopefully shows the cytokine names more clearly.
(7) Please explain the mysterious finding of your study “the increase in beta oxidation activity was not linked to a reduction of plasma or hepatic TAG-levels. In fact, plasma TAG levels were significantly increased in the Papain + Bromelain CPH and the Alcalase CPH group compared to control”. Do you have any references for this, to support your study?
REPLY: We agree that this is an unexpected finding, and have already attempted to address this in the discussion. In the revised version of the manuscript we have specified our previous finding of increased TAG-levels concerning salmon hydrolysates. Discussion, page 22: “Interestingly, we previously observed that one of three salmon hydrolysates generated through different enzymatic procedures increased plasma TAG-levels [20], independent of hepatic b-oxidation activity or feed intake.”
(8) Please update the full information of the cited papers in the references such as volume, issue number and page numbers.
- Anupama N.; Sindhu G.; Raghu K.G. Significance of mitochondria on cardiometabolic syndromes. Fundam Clin Pharmacol 2018.
-- Jensen I.J.; Maehre H.K. Preclinical and Clinical Studies on Antioxidative, Antihypertensive and Cardioprotective Effect of Marine Proteins and Peptides-A Review. Mar Drugs 2016, 14
-- Chai T.T.; Law Y.C.; Wong F.C.; Kim S.K. Enzyme-Assisted Discovery of Antioxidant Peptides from Edible Marine Invertebrates: A Review. Mar Drugs 2017, 15
de Campos Zani S.C.; Wu J.; Chan C.B. Egg and Soy-Derived Peptides and Hydrolysates: A Review of 400 Their Physiological Actions against Diabetes and Obesity. Nutrients 2018, 10
--Vik R.; Parolina C.; Bjorndal B.; Busnelli A.; Holm S.; Brattelid T., et al. A salmon protein hydrolysate excerts lipid-independent anti-atherosclerotic activity in Apo E-deficient mice. Atherosclerosis 2014, 235, E188.
REPLY: We have corrected the references as instructed where possible. The information on page range was unfortunately missing in the journal reference format for four of the references.
After amendment of the above comments in the manuscript, I would support in favor of the author for the publication of this paper.
We hope that the replies to the reviewer’s comments, and the changes made in the text, will make the manuscript acceptable for publication in Medicines.
Yours sincerely,
Bodil Bjørndal
On behalf of the authors
Reviewer 2 Report
The current study was objected to investigate the effect of chicken hydrolyzates on obesity induced inflammation. The results seems to be interesting to this filed researchers. Before acceptance, I suggest to change in a few points.
In figure 3, the authors stated that there was a dramatic decrease in fasting glucose levels, however, there were no marks for statistical analysis. The statistical analysis should be inserted.
The results showed decreasing pattern in proinflammatory cytokines. However, there was no data for normal control group. The discussion should be added.
Author Response
Reply to reviewers:
We greatly appreciate the valued input from the reviewer, and the opportunity to resubmit our manuscript "Chicken protein hydrolysates have anti-inflammatory effects on high-fat diet induced obesity in mice" (Man. no. Medicines-412488). We have attempted to address all the questions of the reviewer, and hope that this has resulted in an improved manuscript. All changes are marked in the revised manuscript.
Reviewer report and replies:
The current study was objected to investigate the effect of chicken hydrolyzates on obesity induced inflammation. The results seems to be interesting to this filed researchers. Before acceptance, I suggest to change in a few points.
In figure 3, the authors stated that there was a dramatic decrease in fasting glucose levels, however, there were no marks for statistical analysis. The statistical analysis should be inserted.
REPLY: We did not include statistics for time-effects of the diets, but as the reviewer points out, this would be of interest to report, and in the revised manuscript we have added marks for statistical analysis in figure 3, and the following paragraphs in the manuscript: Results, Statistical analysis: “Two-way ANOVA with Sidak’s multiple comparisons test was used to evaluate statistical differences within each group at different time points.” New legend to figure 3: “Fasting glucose and insulin levels in C57BL/6 mice at baseline (0 weeks) and after 11 weeks of feeding with 19 E% casein- or 9.5 E% casein and 9.5 E% CPH-high fat/high-sucrose diets. A) The fasting blood glucose level. B) The fasting plasma insulin level. C) The fasting glucose and insulin ratio. Mean values with standard deviation are shown (n = 5-6). Significant difference between group mean values at 0 and 11 weeks were determined by one-way ANOVA with Fisher’s LSD test (all p > 0.05). Significant difference between mean values at 0 and 11 weeks within each group was determined by two-way ANOVA and Sidak’s multiple comparisons test (*p < 0.05, **p < 0.01, ***p < 0.001).”
The results showed decreasing pattern in proinflammatory cytokines. However, there was no data for normal control group. The discussion should be added.
REPLY: We agree that the lack of a low-fat fed control group is a limitation of the study, and this has been mentioned in the discussion: «The lack of a low-fat fed control group is a limitation of the study, and although the cytokine-lowering potential of CPHs was evident, we cannot conclude whether plasma inflammation was returned to normal levels.»
We hope that the replies to the reviewer’s comments, and the changes made in the text, will make the manuscript acceptable for publication in Medicines.
Yours sincerely,
Bodil Bjørndal
On behalf of the authors